# High contiguity *Arabidopsis thaliana* genome assembly with a single nanopore flow cell

Todd P. Michael [1], Florian Jupe[2,6], Felix Bemm[3], S. Timothy Motley[1], Justin P. Sandoval[2], Christa Lanz[3], Olivier Loudet [4], Detlef Weigel [3] & Joseph R. Ecker[2,5]

The handheld Oxford Nanopore MinION sequencer generates ultra-long reads with minimal cost and time requirements, which makes sequencing genomes at the bench feasible. Here, we sequence the gold standard *Arabidopsis thaliana* genome (KBS-Mac-74 accession) on the bench with the MinION sequencer, and assemble the genome using typical consumer computing hardware (4 Cores, 16 Gb RAM) into chromosome arms (62 contigs with an N50 length of 12.3 Mb). We validate the contiguity and quality of the assembly with two independent single-molecule technologies, Bionano optical genome maps and Pacific Biosciences Sequel sequencing. The new *A. thaliana* KBS-Mac-74 genome enables resolution of a quantitative trait locus that had previously been recalcitrant to a Sanger-based BAC sequencing approach. In summary, we demonstrate that even when the purpose is to understand complex structural variation at a single region of the genome, complete genome assembly is becoming the simplest way to achieve this goal.

[1] J. Craig Venter Institute, La Jolla, CA 92037, USA. [2] Genomic Analysis Laboratory, The Salk Institute for Biological Studies, La Jolla, CA 92037, USA. [3] Max Planck Institute for Developmental Biology, 72076 Tübingen, Germany. [4] Institut Jean-Pierre Bourgin, INRA, AgroParisTech, CNRS, Université Paris-Saclay, 78000 Versailles, France. [5] Howard Hughes Medical Institute, The Salk Institute for Biological Studies, La Jolla, CA 92037, USA. [6] Present address: Monsanto Company, Creve Coeur, MO 63141, USA. Todd P. Michael, Florian Jupe, and Felix Bemm contributed equally to this work. Correspondence and requests for materials should be addressed to T.P.M. (email: tmichael@jcvi.org)

The first *Arabidopsis thaliana* genome assembly, from the Columbia (Col-0) reference accession, was released in 2000[1]. Based on Sanger sequencing of a BAC tiling path, and subsequent improvements, this reference assembly has become a "gold standard" not only for this species but for eukaryotic genomics in general. Nevertheless, the latest version still contains 29 larger mis-assemblies[2], has 117 gaps with unknown bases ('Ns'), and is missing about 25 Mb of repeat (mostly centromere) sequence[3]. The advent of high-density microarrays and later Illumina short-read sequencing led to several efforts to access the nucleotide and structural diversity of genomes from other *A. thaliana* accessions[3–7]. Most of these efforts were short read and reference based, revealing only limited structural changes. Various attempts at reference guided or de novo assembly with short reads have been pursued[8,9], but these also suffered from being unable to accurately reconstruct larger insertions as well as presence/absence polymorphisms of medium-copy repeats.

The read lengths of third-generation sequencing technologies, such as SMRT Sequencing from Pacific Biosciences (PacBio), exceed those of major repeats in many genomes, and are now enabling highly contiguous genome assemblies[10,11]. Highly repetitive regions such as the ribosomal DNA (rDNA) and centromeres, however, still remain mostly unassembled with third-generation technologies due to limitations in read length and error rate. Oxford Nanopore Technology's (ONT) MinION sequencing has overcome at least one of these challenges by producing reads that exceed 200 kb, but these still have 5−15% per-base error rates (R7.3; ref. [12]). While ONT long reads have primarily been used to sequence smaller bacterial genomes, recently the 860 Mb European Eel (*Anguilla anguilla*) and 1.2 Gb wild tomato (*Solanum pennellii*) genomes were updated using ONT plus Illumina and ONT only strategies, respectively, resulting in highly contiguous assemblies[13,14]. Optical genome mapping using the Bionano Genomics system, on the other hand, allows long-range scaffolding and identification of large genome variation and mis-assemblies. To resolve the underlying genome sequence, it needs to be combined with long-read sequencing[2].

As sequencing costs are declining, it has become apparent that even when the goal is only a single gene, or single genomic region, whole-genome or whole-chromosome assemblies are often the fastest way to reconstruct complex local sequences[15]. Unfortunately, for most efforts, it has been difficult to justify the associated costs. Here, we show how one can rapidly and inexpensively resolve structural variation (SV) at a quantitative trait locus (QTL) in *A. thaliana* by taking advantage of a highly contiguous genome assembly produced from 2 μg genomic DNA on just a single ONT MinION flow cell (R9.4). We validated the contiguity and the quality of this assembly with Bionano optical maps and a second assembly generated from reads produced on the PacBio Sequel platform. The final assembly, which was generated on a laptop in only 4 days, covers 100% of the non-repetitive genome space, with only a fraction of the gaps present in the current gold standard Arabidopsis TAIR10 assembly. We show the utility of such a rapidly created genome by identifying an intrachromosomal duplication responsible for a growth phenotype that could not be resolved with a Sanger-based BAC sequencing approach.

## Results

**MinION sequencing of *Arabidopsis thaliana* KBS-Mac-74.** We extracted high-molecular weight DNA from the leaf tissue of the *A. thaliana* accession KBS-Mac-74 (1001 Genomes accession 1741), and used this DNA to prepare a library with the ONT

Ligation Sequencing Kit 1D (SQK-LSK108). Limited shearing is an inevitable result of DNA purification. This minimally sheared DNA, instead of intentionally sheared and size selected DNA, is our standard protocol to access ultra-long reads. The DNA was sequenced using a single ONT MinION R9.4 flowcell (FLO-MIN106) for 48 h. The resulting 300,053 fast5 files (sequencing reads) were subsequently analyzed using the Albacore recurrent neural network (RNN) base-caller (v0.8.4). The average read length was 11.4 kb (N50 7.5 kb), and generated a total of 3.4 Gb of base-called sequence. Oxford Nanopore is rapidly improving the Albacore base-caller and subsequent versions (up to v2.1.3 in December 2017) provided only minimal quality and quantity improvements. The modest improvements are consistent with other ONT plant sequencing projects[14] and may reflect that the Albacore RNN was not trained on unamplified plant DNA.

In the pool of KBS-Mac-74 reads, there were four (4) reads longer than 200 kb, including one of 269 kb, 14 longer than 100 kb, and 2317 longer than 50 kb. The reads had an average quality of Q7.3. Plotting GC content against read length revealed that the lower quality reads had a slightly larger GC spread, and some of the longest reads were skewed toward low GC (25%), compared to an average of 36% in the *A. thaliana* reference genome (Fig. 1a, c). When we assessed the reads by mapping, we found that the low quality, GC-spread reads dropped out (Fig. 1a versus b), as did the low GC long reads (Fig. 1c versus d).

**Genome assemblies and inter-platform comparison.** While several long-read, read error-tolerant assemblers, such as Canu[16] and Falcon[17], produce high quality genomes, they are compute intensive for larger eukaryotic genomes. In contrast, minimap/miniasm is an assembler specifically designed to handle long error-prone reads in a fraction of the time[18]. We leveraged Canu and minimap/miniasm to assemble raw ONT fastq reads. Quality and contiguity of the two ONT assemblies (Canu: ONTcan; minimap/miniasm: ONTmin) were compared with a "state-of-the-art" assembly generated by Falcon[17] from Pacific Biosciences (PacBio) long reads (PBfal; Supplementary Fig. 1), as well as Bionano Genomics optical genome maps. Both ONT assemblies were comparable to PBfal as deduced from common assembly metrics and showed a high contiguity (Table 1). Of note, the ONTmin assembly was the smallest (110.9 Mb), but had the fewest contigs (62), the second longest N50 (11.5 Mb) and longest single contig (13.8 Mb).

The minimap/miniasm assembly strategy does not use initial error correction of raw reads as implemented in the Canu pipelines, and therefore the raw assembly suffers from a lower per-base accuracy (Table 2). We polished the assemblies using three (3) rounds of racon[19], followed by one round of polishing with pilon[20] using 30× Illumina PCR-free paired-end reads (2 × 250 bp) (Table 2). In general, racon significantly improved the total lengths of ONT assemblies (Fig. 2a), and increased N50 length to 12.3 Mb and longest contig to 14.8 Mb (Table 1). After three rounds of racon and one round of pilon, the global quality (see Methods, section Genome assembly validation and comparison) of the ONTmin assembly was at Q32 and thus of lower quality than the PBfal assembly (Q37) and the TAIR10 reference genome (Q39). Only the first round of racon polishing and the final pilon polishing had strong effects on single-base mismatches (false-SNPs), false-insertions, and false-deletion corrections (Fig. 2, Table 2).

**Quality control of base and structural quality with optical genome maps.** Bionano Genomics (Bionano) optical genome maps have been useful in identifying mis-joins and assessing the contiguity of short- and long-read assemblies[2,21–23]. We

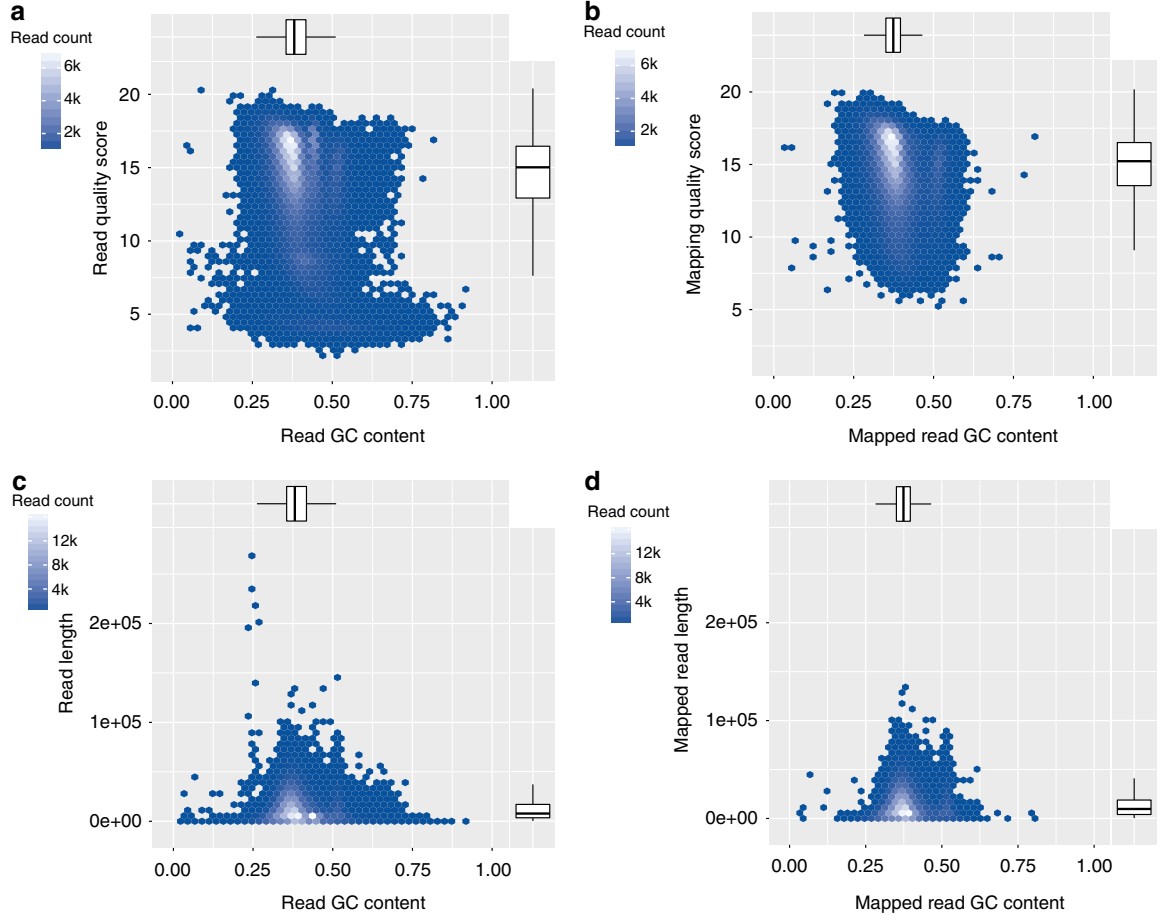

**Fig. 1** *Arabidopsis thaliana* Oxford Nanopore MinION read quality. **a** Read quality versus GC content. **b** Aligned read quality versus GC content of mapped reads. **c** Read length versus GC content. **d** Aligned read length versus GC content of mapped reads. Lower and upper hinges correspond to the 25th and the 75th percentile. Whiskers are extended 1.5 × IQR (IQR is the interquartile range between the 25th and the 75th percentile) from the smallest and highest hinge

**Table 1 Metrics of the KBS-Mac-74 Oxford Nanopore MinION and Pacific Biosciences Sequel assemblies**

| Name | Round (#) | Contig (#) | Total length (bp) | N50 length (bp) | Longest (bp) |
|---|---|---|---|---|---|
| ONTcan | 0 | 82 | 113,402,589 | 8,491,723 | 13,170,991 |
| ONTcan | 1 | 82 | 115,632,969 | 8,666,702 | 13,418,669 |
| ONTcan | 2 | 82 | 115,735,933 | 8,671,757 | 13,431,826 |
| ONTcan | 3 | 82 | 115,781,606 | 8,673,416 | 13,434,963 |
| ONTcan | 4 | 82 | 117,857,778 | 8,825,005 | 13,671,933 |
| ONTmin | 0 | 62 | 110,880,468 | 11,450,026 | 13,809,770 |
| ONTmin | 1 | 62 | 117,109,669 | 12,091,692 | 14,572,214 |
| ONTmin | 2 | 62 | 117,377,831 | 12,118,333 | 14,600,216 |
| ONTmin | 3 | 62 | 117,441,618 | 12,123,124 | 14,605,848 |
| ONTmin | 4 | 62 | 119,502,799 | 12,334,794 | 14,864,299 |
| PBfal | 0 | 78 | 119,340,317 | 10,700,858 | 13,315,705 |
| PBfal | 1 | 78 | 119,755,332 | 10,737,485 | 13,359,771 |
| PBfal | 4 | 78 | 119,750,280 | 10,736,982 | 13,359,931 |

Assembly types: Oxford Nanopore MinION, ONT; PacBio Sequel, PB; miniasm, min; Falcon, fal; Canu, can. Polishing rounds PBfal: 0 = raw assembly; 1, arrow 1x; 4, pilon 1x

**Table 2 Variant calling metrics of Illumina PCR-free paired-end reads against the assemblies and Illumina PCR-free paired-end reads from Col-0 mapped against the TAIR10 reference assembly**

| Name | Round (#) | Total variants (#) | SNPs (#) | Insertions (#) | Deletions (#) |
|---|---|---|---|---|---|
| ONTcan | 0 | 2,505,080 | 281,421 | 2,202,363 | 21,296 |
| ONTcan | 1 | 1,782,042 | 276,025 | 1,432,415 | 73,602 |
| ONTcan | 2 | 1,723,244 | 266,625 | 1,384,435 | 72,184 |
| ONTcan | 3 | 1,706,676 | 264,720 | 1,369,595 | 72,361 |
| ONTcan | 4 | 99,252 | 73,616 | 10,244 | 15,392 |
| ONTmin | 0 | 7,208,897 | 3,359,123 | 2,911,987 | 937,787 |
| ONTmin | 1 | 1,994,370 | 345,671 | 1,532,261 | 116,438 |
| ONTmin | 2 | 1,767,580 | 287,706 | 1,393,680 | 86,194 |
| ONTmin | 3 | 1,730,743 | 278,301 | 1,369,470 | 82,972 |
| ONTmin | 4 | 91,544 | 67,062 | 9026 | 15,456 |
| PBfal | 0 | 474,355 | 62,063 | 367,338 | 44,954 |
| PBfal | 1 | 39,433 | 26,486 | 6860 | 6087 |
| PBfal | 4 | 27,312 | 24,744 | 1010 | 1558 |
| TAIR10 | – | 16,807 | 13,336 | 1770 | 1121 |

Assembly types: Oxford Nanopore MinION, ONT; PacBio Sequel, PacBio; Miniasm, min; Falcon, fal; Canu, can; reference genome, TAIR10. Polishing rounds PBfal: 0 = raw assembly; 1, arrow 1x; 4, pilon 1x

generated 265 optical genome maps with contig (cmap) lengths up to 4.8 Mb and an N50 of 695 kb. We used these to assess the quality of the ONT sequence assemblies by screening for chimeric contigs and mis-assemblies such as collapsed and artificially expanded regions (Supplementary Table 1). While we found only

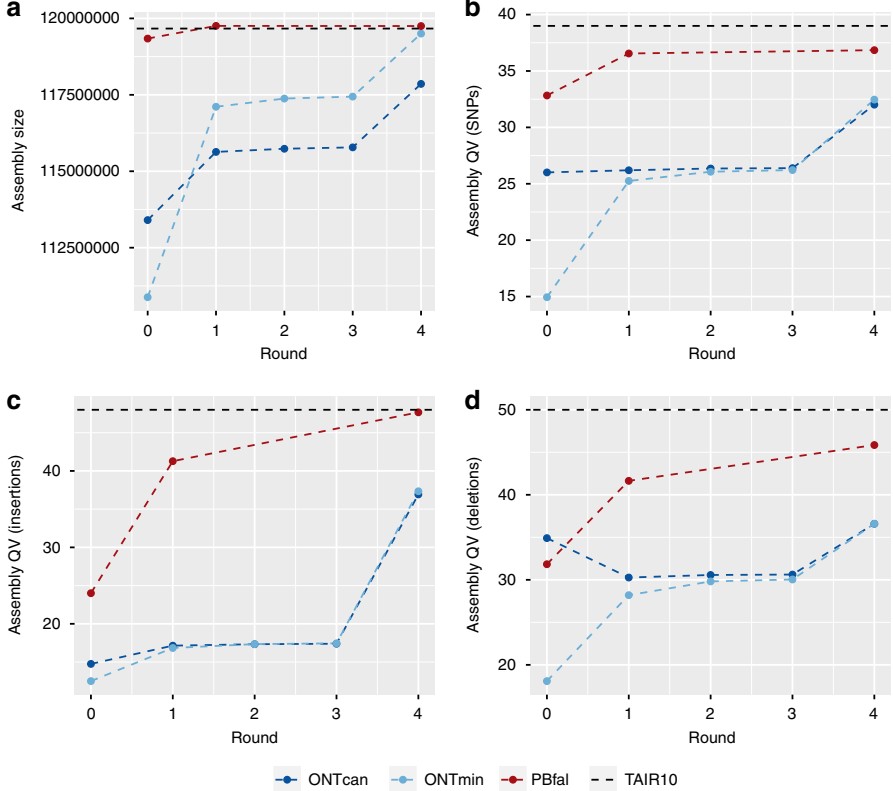

**Fig. 2** Quality comparisons of single molecule assemblies and TAIR10 reference assembly. **a** Final assembled genome size versus assembly rounds. **b** Assembly quality based on short sequence artifacts versus rounds. **c** Assembly quality based false-insertions versus rounds. **d** Assembly quality based false-deletions versus rounds. Assembly types: Oxford Nanopore MinION, ONT; PacBio Sequel, PB; miniasm, min; Falcon, fal; Canu, can; reference genome, TAIR10. Polishing rounds PacioBfal: 0 = raw assembly; 1, arrow 1x; 4, pilon 1x

a single chimeric contig in ONTcan, all other approaches assembled this particular region perfectly (Fig. 3a, b). Looking at the final round 4 of each assembly, we found three artificially expanded regions in ONTmin, none in the ONTcan, and a single artificial expansion in the PBfal assembled contigs (Supplementary Table 1). One example of an erroneously extended region of ~18 kb, most likely a repeat, within ONTmin is shown in Fig. 3d. Within the final ONTmin contigs, we identified six collapsed sequences within highly repetitive regions, totaling 244,011 bp (Supplementary Table 1; Fig. 3c).

We further measured global base quality through in silico labeling of the contigs at Nt.BspQI sites and comparison to Bionano maps, also based on Nt.BspQI restriction sites. This analysis revealed false-positive (FP) and false-negative (FN) Nt. BspQI sites, which we used to compare the five rounds per assembly against the KBS-Mac-74 Bionano maps (Supplementary Table 1). The raw ONTmin assembly (round 0) failed to produce good alignments with optical maps due to high nucleotide errors, causing little overlapping labels. The dramatic increase in base quality after racon polishing, however, already resulted in Bionano map alignment with 115.9 Mb of contigs, which increased to 118.4 Mb in the final ONTmin assembly (round: 4). FP/FN ratios decreased from 0.33/0.08 to 0.02/0.04, with another dramatic shift due to the final pilon polishing step (round: 4). While changes were marginal between racon rounds 2 and 3, the pilon polishing step improves both assembly types, decreasing the FP/FN ratios in ONTcan (round: 4) from 0.33/0.12 to 0.01/0.04 (Supplementary Table 1, Fig. 2).

**Limited assembly of ultra-long repetitive regions.** Because minimap/miniasm does not attempt to assemble repeat

sequences[18], we determined how the three assemblies differed in repeat content. While we were not able to identify large repeats, such as the nucleolar organizer rDNA (NOR) on the short arms of chromosome 2 and 4, the assemblies contained telomeric and centromeric repeats. Examples include a 2.8 kb telomere repeat on contig 1 (Fig. 3e) and two centromeric repeat arrays of 16.5 and 4.8 kb at the end of contig 3 (Fig. 3f). The PBfal assembly had a similar amount of assembled centromere and rDNA sequences. In addition, repeat calling with RepeatMasker[24] revealed a similar number of repetitive elements in the assemblies, and de novo LTR prediction identified a similar number (~700) of full length elements. As an additional proxy for the quality of the ONTmin assembly, we screened for complete and partial genes based on a whole-genome alignment against the TAIR10 reference genome using Quast. Similar gene content was predicted for each assembly. The ONTmin assembly had the highest completeness, containing all Araport11 predicted protein coding genes[25], closely followed by the ONTcan assembly (Supplementary Table 2). Overall, polishing brought the ONT assemblies to a comparable level in base quality and contiguity to the state-of-the-art PacBio assembly of the same genome.

**Chromosome arm contig reveals trait gene and QTL structure.** The main purpose of demonstrating that highly contiguous genome assemblies can be produced "at the bench" is to resolve SV at a specific QTL. As background, we compared our ONTmin directly with the *A. thaliana* reference genome (TAIR10) to detect SVs. The ONTmin assembly had 4280 SVs with a total length of 9.5 Mb (Supplementary Table 3; Supplementary Fig. 2). Repeat contraction or expansion made up the majority of SV sequences (58%), followed by insertions and deletions (31%). A minority of

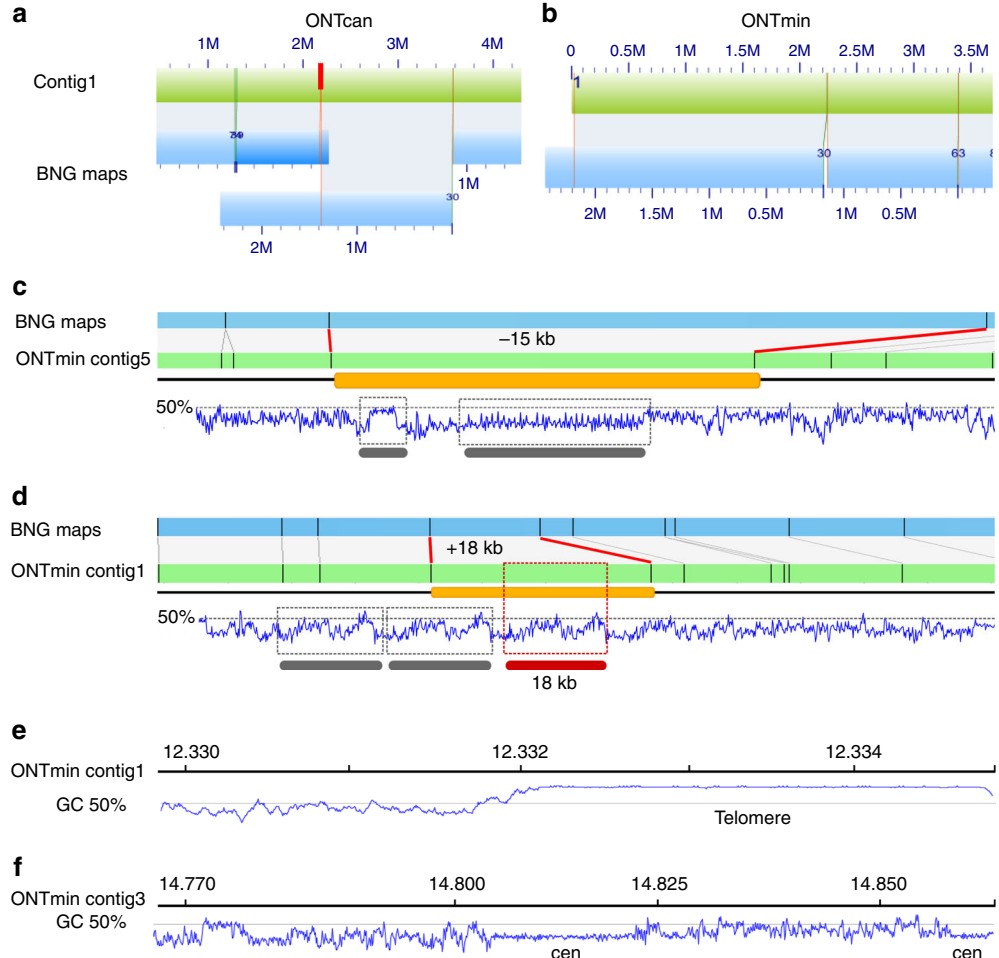

**Fig. 3** Bionano Genomics maps identify mis-assemblies and hard to assemble regions in the Oxford Nanopore MinION assembly. BNG cmap_30 (blue; marked as 30) identified **a** a chimeric ONTcan contig 1 (green) and **b** the correct assembled contig 1 in the ONTmin assembly (green). The chimeric position is indicated with a red bar. **c** A collapsed region in ONTmin contig 5, in which approximately 15 kb sequences are missing from one of the two potential repeat regions as identified by the GC pattern (gray bars). In contrast, **d** shows a falsely duplicated region of approximately 18 kb, with the duplicated repeat region highlighted (red bar, 18 kb). **e** ONTmin assembly resolves various telomere regions, for example after 12.332 Mb on contig 1, as outlined by a GC plot (blue line). **f** ONTmin also resolves short centromere arrays as shown toward the end of contig 3 (blue, GC plot)

SVs were classified as tandem contraction or expansion (11%). Insertion and deletions, as well as tandem contractions and expansions, ranged from 0.5 to 10 kb. Only repeat contractions were longer and usually between 10 and 50 kb. In conjunction with the amount of heterozygous SNPs (Supplementary Fig. 3), which are not expected for a selfing species such as *A. thaliana*, it is likely that these repeat contractions indicate assembly artifacts (Fig. 3c, d).

A major goal of our work was to support the analysis of two interacting QTL, SG3 and SG3i. SG3 had already been fine-mapped to a 9-kb region around the At4g30720 gene[26]. The *A. thaliana* Bur-0 accession has an additional copy of this gene, about 10 Mb away on the same chromosome, and this defines the SG3i QTL. Unfortunately, it had been impossible to exactly define the SG3i region even with Sanger-based BAC sequencing[26]. To analyze this QTL region in KBS-Mac-74, without prior genetic knowledge, we conducted a blast search with the Col-0 At4g30720 genomic sequence against our polished KBS-Mac-74 ONTmin assembly (from round 4). We recovered two hits (100% and 98% identity, 100% coverage) that were 10 Mb apart on the same contig, consistent with the genetic structure proposed for Bur-0 (Fig. 4). We then asked if the breakpoints of the SG3i locus (absent from TAIR10) could be defined by aligning the SG3i region of our polished ONTmin assembly to the Col-0 reference genome (TAIR10). We mapped Col-0 left and right breakpoints at 5,771,424 and 5,773,387 bp, respectively, spanning 1963 bp and containing a predicted transposable element with reverse transcriptase homology (At4g08995 in Col-0). The corresponding KBS-Mac-74 region was much larger and spanned 39,273 bp, containing a paralog of At4g30720. Bur-0 and KBS-Mac-74 are thus very similar, containing about 37 kb extra sequence missing from the Col-0 reference accession. More detailed analysis of the KBS-Mac-74 SG3i region revealed that it contained four fragments of the transposable element consistent with several rounds of nested transposition, which partially explains why it was impossible to amplify this sequence from a BAC clone using PCR[26].

## Discussion

We have shown an important advance for next-generation sequencing. A reference grade eukaryotic genome assembly can be generated in a standard lab environment, and may now represent the fastest path to resolving regions from the reference genome, even in cases where the target is only one such region. In one ONT MinION run, we sequenced and assembled an *A.*

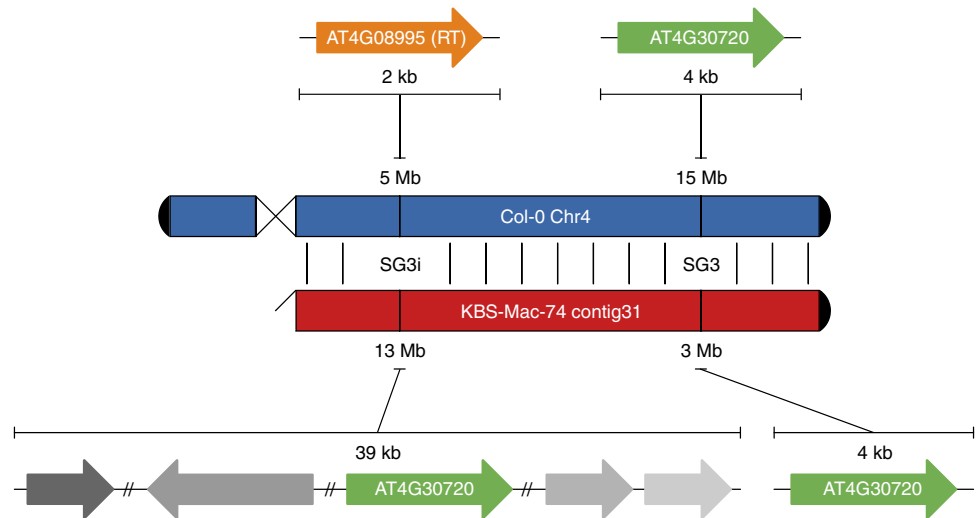

**Fig. 4** Resolution of the At4g30720 duplication using the KBS-Mac-74 ONTmin genome. Col-0 only has one copy of At4g30720 on the distal end (15 Mb) of chromosome 4 (Chr4). KBS-Mac-74 (ONTmin) assembly has two copies of At4g30720; one at the beginning of contig 31 that corresponds to the SG3 QTL location, and one at the distal end of contig 31 that overlaps with the SG3 interacting (SG3i) region in the middle of chromosome 4. At the SG3i locus, the Col-0 gene At4g08995 (889 bp), which is annotated as a transposable element (RT; putative reverse transcriptase), is replaced in the KBS-Mac-74 ONTmin assembly with a 39-kb expansion that includes a duplicated copy of At4g30720. Fragments of the transposable element (different gray arrows) are scattered across the KBS-Mac-74 region consistent with several rounds of transposition resulting in this complex rearranged region

*thaliana* accession (KBS-Mac-74) to a higher contiguity (62 versus 94 contigs[1]) and at comparable base quality with the current gold standard TAIR10 assembly of the Col-0 reference accession. Comparisons to the Col-0 reference genome identified 9.52 Mb affected by structural variants with up to 90 kb of length. While the initial base quality of the minimap/miniasm assembly (ONTmin) was lower than the unpolished Canu assembly (ONTcan), several rounds of racon and one round of pilon applied to both assemblies converged their quality. The quality of the state-of-the-art Pacbio assembly (PBfal) was still slightly better (ONTmin = Q32; PBfal = Q37; Fig. 2; Table 2). Of note, the same trimmed Illumina data set was used for pilon polishing and the quality assessment, thus the latter might be biased by the use of the same data.

The 1D ONT (R9.4) reads generated in this study sporadically surpassed the PacBio Sequel reads in length (Supplementary Fig. 1a, b). However, after read mapping analysis, the longest ONT reads, which were skewed for GC content, dropped out (Fig. 1c, d), as did the lowest-quality reads (Fig. 1a, b). We have generated mappable reads up to 800 kb on MinION flowcells (R9.4) and reads exceeding 1 Mb have been reported by Oxford Nanopore users. Our library protocol that leverages purified genomic DNA without additional shearing, increases the potential to capture longer reads, coupled with the fact that the MinION does not preferentially sequence shorter reads, may have influenced our highly contiguous assembly. These extremely long reads provide a completely new opportunity to assess also highly nested transposable elements and repeat regions that until now were inaccessible to DNA sequencing technologies. Nevertheless, these reads did not enable us to resolve centromeres and rDNA arrays, which have close to 100% sequence identity span multiple Mb in *A. thaliana*, which exceeds the quality and length of even our best ONT data sets.

While we have had great success with Canu[11] for state-of-the-art PacBio assemblies[11], the assembly of ONT reads took several days (5–10 days, depending on the data set), as compared to the minimap/miniasm raw assemblies, which usually took less than 6 h on typical consumer computing hardware with 4 Cores and

16 Gb RAM. The trade-off is that the minimap/miniasm assemblies have substantially lower per-base quality than the Canu assemblies. Minimap/miniasm takes a "correction-free" assembly approach[18], which works particularly well with exceptionally long ONT, error-prone reads. Not only did several rounds of racon improve the mismatch, false-deletion, and false-insertion rate, it also properly extended the ONT assemblies to 100% of the non-repetitive genome size (Fig. 2a). Most importantly, a final pilon polish with Illumina PCR-free reads brought all of the assemblies to a similar base quality level as observed by variant detection, and protein-coding gene as well as transposable element prediction.

As an independent validation of our assemblies, we used Bionano optical genome maps to test the assembly quality and contiguity[23]. Bionano maps are now routinely used to identify structural genome variations[2,23], and we have expanded their utility by using the inherent long-range physical/optical information to screen for chimeric sequencing contigs, collapsed and artificially expanded regions larger than 5 kb (Fig. 3a–c). These errors occur in highly repetitive regions, as previously described for Col-0[2], yet within long-read assemblies at a much lower rate compared to the BAC-based TAIR10 reference. In genome assembly projects, contigs are scaffolded through secondary, overlapping datasets, such as optical maps[22] and Hi-C data[23]. To facilitate the best possible matches, the assembled contigs need to be polished to highest quality levels. Here, we exploited Nt.BspQI nicking as a proxy for base-quality through the assessment of FP sites that were nicked even though they are different from the recognition sequence of the restriction enzyme used, or FN nicking sites (sites that had the recognition sequence but were not nicked). Through our comparisons, we have observed drastic improvements in FP and FN values of Nt.BspQI recognition sequence (5′-GCTCTTCN^-3′) within contigs. Especially the reduction of FN sites between individual polishing rounds led to an increase in contigs that can be aligned to optical maps. Base quality is thus an important factor not only for downstream genome annotations, but also for scaffolding with secondary data sets. Once again, Bionano maps confirmed that the base quality

and contiguity of the ONT assemblies were similar to or even surpassed that of the current version of the Col-0 reference genome.

Ultra-long read reference genomes present new opportunities to access biologically relevant variation that is even recalcitrant to the most trusted genomic resources. Despite having had access to Sanger-sequenced BACs from the SG3i QTL region, we were previously unable to fully resolve the complex genomic structure of this region[26]. With the help of the polished KBS-Mac-74 ONTmin assembly, we could dissect the SG3i region in less than 30 min. Recently, the 100 Mb model *Caenorhabditis elegans* genome was de novo assembled with ONT MinION to resolve complex rearrangements in a mutagenized strain, which had previously been unresolvable by other techniques[27]. Coupled to the fact that this assembly could be completed at the bench in days, now researchers can directly address complex genomic regions instead of using more laborious targeted approaches.

In conclusion, we have demonstrated that highly contiguous genome assemblies with high per-base quality have the potential to resolve functionally important presence/absence polymorphisms in *A. thaliana* accessions, and that whole-genome assemblies will soon become the standard method to approach even individual complex regions of the genome[15].

## Methods

**Plant growth**. *A. thaliana* accession KBS-Mac-74 (accession 1741) was grown in the greenhouse under 16 h light and 8 h dark at 22 °C. Plants were put into complete dark for 48 h before DNA extraction, in order to reduce starch accumulation.

**Oxford Nanopore MinION sequencing**. Five grams of flash frozen leaf tissue was ground in liquid nitrogen and extracted with 20 mL CTAB/Carlson lysis buffer (100 mM Tris-HCl, 2% CTAB, 1.4 M NaCl, 20 mM EDTA, pH 8.0) containing 20 μg/mL proteinase K for 20 min at 55 °C. The DNA was purified by addition of 0.5× volume chloroform, which was mixed by inversion and centrifuged for 30 min at 3000 RCF, and followed by a 1× volume 1:1 phenol: [24:1 chloroform:isoamyl alcohol] extraction. The DNA was further purified by ethanol precipitation (1/10 volume 3 M sodium acetate pH 5.3, 2.5 volumes 100% ethanol) for 30 min on ice. The resulting pellet was washed with freshly prepared ice-cold 70% ethanol, dried, and resuspended in 350 μL 1× TE buffer (10 mM Tris-HCl, 1 mM EDTA, pH 8.0) with 5 μL RNase A (Qiagen, Hilden) at 37 °C for 30 min, followed by incubation at 4 °C overnight. The RNase A was removed by double extraction with 24:1 chloroform:isoamyl alcohol, centrifuging at 22,600×g for 20 min at 4 °C each time. An ethanol precipitation was performed as before for 3 h at 4 °C. The pellet was washed as before and resuspended overnight in 350 μL 1× TE.

Genomic DNA sample was further purified for ONT sequencing with the Zymo Genomic DNA Clean and Concentrator-10 column (Zymo Research, Irvine, CA). The purified DNA was then prepared for sequencing following the protocol in the genomic sequencing kit SQK-LSK108 (ONT, Oxford, UK). Briefly, approximately 2 μg of purified DNA, without exogenous shearing or size selection, was repaired with NEBNext FFPE Repair Mix for 60 min at 20 °C. The DNA was purified with 0.5× Ampure XP beads (Beckman Coulter). The repaired DNA was End Prepped with NEBNExt Ultra II End-repair/dA tail module and purified with 0.5× Ampure XP beads. Adapter mix (ONT, Oxford, UK) was added to the purified DNA along with Blunt/TA Ligase Master Mix (NEB, Beverly, MA) and incubated at 20 °C for 30 min followed by 10 min at 65 °C. Ampure XP beads and ABB wash buffer (ONT, Oxford, UK) were used to purify the library molecules and they were recovered in Elution buffer (ONT, Oxford, UK). Purified library was combined with RBF (ONT, Oxford, UK) and Library Loading Beads (ONT, Oxford, UK) and loaded onto a primed R9.4 Spot-On Flow cell (FLO-MIN106). Sequencing was performed with a MinION Mk1B sequencer running for 48 h. Resulting FAST5 files were base-called using the Oxford Nanopore Albacore software (v0.8.4) using parameters for FLO-MIN106, and SQK-LSK108 library type. Oxford Nanopore is rapidly developing the Albacore RNN base-caller and several versions of Albacore were released during the review of this work (v0.8.4 to v2.1.3). We tested v1.2.1, v1.2.4, v2.0.2, and v2.1.3 and found that the quality and quantity of sequence called was only marginally better than v0.8.4, consistent with another report on base-calling unamplified plant DNA[14].

**Illumina sequencing**. Genomic DNA was isolated from the leaf tissue for both KBS-Mac-74 and Col-0 using Qiagen DNeasy Plant Mini Kit. Paired-end libraries (PE) were prepared from gDNA sheared to ~500 bp using an S2 Focused-Ultrasonicator (Covaris Inc., MA, USA), and the TruSeq DNA PCR-Free Library Preparation Kit (Illumina, Inc., San Diego, US-CA). PE libraries were sequenced on

an Illumina MiSeq using a 500-cycle MiSeq Reagent Kit v2. Resulting reads were trimmed and filtered for high quality (Q38). Only paired reads were used for pilon polishing.

**Oxford Nanopore MinION assembly and correction**. Raw ONT reads (fastq) were extracted from base-called FAST5 files using poretools[28]. Overlaps were generated using minimap[18] with the recommended parameters (-Sw5 -L100 -m0). Genome assembly graphs (GFA) were generated using miniasm[18]. Unitig sequences were extracted from GFA files. Canu assemblies were generated using the default parameters and the complete canu pipeline[16]. Three rounds of consensus correction was performed using racon[19] based on minimap overlaps, and the resulting assembly was polished using Illumina PCR-free 2 × 250 bp reads mapped with bwa[29] and pilon[20]. Genome stats were generated using QUAST[30]. The resulting assembly was in 62 contigs with an N50 of 12.3 Mb.

**Preparation of Bionano optical genome maps**. High-molecular weight (HMW) DNA was extracted from up to 5 g fresh leaf tissue using a modified Bionano Genomics protocol[2]. Briefly, extracted HMW DNA was nicked with the enzyme Nt.BspQI (NEB, Beverly, MA, USA), fluorescently labeled, repaired, and stained overnight according to the Bionano Genomics nick-labeling protocol[31]. KBS-Mac-74 nick-labeled DNA was run on a single flow cell on the Irys platform (Bionano Genomics, San Diego, CA, USA), for 90 cycles to generate 22.5 Gb raw data. The IrysView software (Bionano Genomics; version 2.5.1) was used to quality filter the raw data (>100 kb length, >2.75 signal/noise ratio) and molecules were assembled into contigs using the default "human genome" parameters. Resulting Bionano cmaps were compared against the different assemblies using Bionano RefAlign[2], and collapsed regions or artificial expansions were detected as structural variations using the structomeIndel.py script (https://github.com/RyanONeil/structome). Rates for FP and FN nicking sites were extracted from the .err files, and genome coverage was computed from the .xmap files.

**Genome assembly validation and comparison**. Genome assemblies were validated with a variant calling based approach. Illumina short reads were mapped against each assembly with bwa[29] and variations called with FreeBayes[32]. Genome-wide quality values (Q) for SNPs, insertions, and deletions were calculated as $Q = -10 \times \log_{10}$(total length of variants / total length of sites (DP >3))[33,34]. Only biallelic variants were considered for quality validation. Assemblies were repeat masked using RepeatMasker[24]. KBS-Mac-74 specific centromere (158 and 178 bp monomers) and rDNA (26S, 5.8S, 18S, and 5S) repeats were identified in the PBfal assembly with blast (blastn) using representative sequence from Col-0 TAIR10; the KBS-Mac-74 specific repeats were then used to search all four assemblies. Structural variation between assembled genomes and TAIR10 were detected with Assemblytics[35] based on whole genomes alignments generated with Mummer[36]. For SG3/SG3i analysis, the complete genomic region for At4g30720 was blasted against the KBS-Mac-74 ONTmin round 4 assembly; 50 kb upstream and downstream of the two At4g30720 KBS-Mac-74 ONTmin hits were used to align to the Col-0 TAIR10 assembly to identify insertions, deletions, and rearrangement.

**Data availability**. Raw sequencing data were deposited in the European Nucleotide Archive (ENA) under project PRJEB21270. Raw Bionano Genomics molecules and assembled maps are deposited under BioProject ID PRJNA390205. Final polished assemblies were deposited in the ENA Genome Assembly Database: PacBio Sequel, PRJEB23084 and Oxford Nanopore MinION, PRJEB21270. Code snippets and further data were deposited at GitHub (https://github.com/fbemm/onefc-oneasm).

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

## Acknowledgements

This work was funded by ERC AdG IMMUNEMESIS and the Max Planck Society. F.J. was supported by a Human Frontier Science Program Organization long-term fellowship. J.R.E. is an investigator of the Howard Hughes Medical Institute.

## Author contributions

T.P.M., F.J., F.B., D.W., and J.R.E. conceived the study, analyzed the data, and wrote the manuscript. S.T.M., C.L., and F.B. generated the ONT, PacBio, and Illumina sequence, respectively. F.J. and J.P.S. generated and analyzed the BNG data. O.L. and T.P.M. analyzed the SG3/SG3i QTL.
