## [Peer Review File · Nature Communications]

Reviewers' comments:

Reviewer #1 (Remarks to the Author):

The article (Manuscript#: NCOMMS-17-15663-T) entitled "High contiguity Arabidopsis thaliana genome assembly with a single nanopore flow cell", by Todd Michael and collaborators delivers "a fast and cost effective reference assembly for an Arabidopsis thaliana accession using the USB-sized Oxford Nanopore MinION sequencer and typical consumer computing hardware (4 Cores, 16Gb RAM)", aiming at reducing sequencing and computing cost for de novo genome assembly. While the work looks interesting, the methodological/biological insights provided here are of limited novelty. Specifically, the authors sequenced an A. thaliana accession using the Oxford Nanopore MinION sequencer, and used miniasm/racon pipeline to create assemblies that are comparable to CANU assemblies but at lower computing cost. None of the technologies used here were novel and the authors basically provided testing results for these existing technologies using a newly sequenced A. thaliana genome, which however provided very limited biological insights. In particular, the miniasm/racon pipeline has been shown to work well with C. elegans PacBio data (Sović et al. 2017. Genome Res. doi:10.1101/gr.214270.116), which has comparable genome complexity with A. thaliana. Although the testing by Sović et al. was performed using PacBio data, the MinION data provided here were of similar quality (similar average length, except some longer reads, and similar error rate), which should lead to comparable results by the same assembly pipeline (and indeed the results of the two genomes generated by miniasm/racon looked similar or comparable in assembly statistics).

Below are some specific concerns regarding the current study.

In Abstract, the authors stated that "because these technologies are not only costly, but also time and compute intensive, it has been unclear how scalable they are". This is misleading. PacBio SMRT technology has been successfully used in assembling large plant genomes such as maize, sunflower, and also partially helped wheat genome assembly. Even though these assemblies required a large amount of computing resource, the scalability of these methods has been proved. Even though cost is still a big issue, it is not unacceptable. Please note that genome assembly, even with the reduced cost, is still not intended to be routinely performed in every lab, especially considering the amount of follow-up work such as genome annotations that is required to make the genome sequence useful. Furthermore, the Arabidopsis genome is just a 'tiny' genome in the plant kingdom (and also very small in animal world). A method that is successful for Arabidopsis doesn't automatically extend to large and complex plant genomes. For example, in our preliminary testing with the recently published maize B73 PacBio data, miniasm clashed on a computer with 1 TB memory. To test the scalability of the methods presented here, the authors should at least use a published SMRT dataset of complex genomes such as maize to make sure that the methods are indeed able to work as expected.

Reviewer #2 (Remarks to the Author):

This research represents a substantive advance in the field of genomic sequencing, in particular demonstrating that high quality assemblies of a higher plant genome can be achieved using a low-cost, long-read technology, and light-weight computational tools. The work demonstrates that the main weakness of long-read technologies, low base accuracy, need not require computationally expensive read-correction steps prior to assembly. Rather, low accuracy can be overcome using consensus correction and polishing steps performed after primary assembly, as claimed by the developers of these tools in prior publications, but demonstrated independently here. By comparing Oxford Nanopore technologies with PacBio, this work also shows that comparable quality can be achieved but

at much lower cost. This work also serves as a guiding example of how genome assemblies can and should be evaluated using optical map comparison and short-read mapping. This work is anticipated to have high impact on the research community as a "methods" paper. Although this work provides a use-case to demonstrate the utility of the resulting Arabidopsis strain KBS-Mac-74 reference assembly in detecting the likely genetic basis of a QTL, it falls short of any comprehensive comparative analysis with the Col-0 strain upon which TAIR10 is based. The latter analysis would increase the intellectual merit of this work, but perhaps is being reserved for a separate publication by these authors.

The manuscript should be improved in the following ways before publication:

- 1) More precise terms to describe sequence/assembly artifacts so as to not confuse with real genetic variation:
 - a. For example, in Table 2 and elsewhere the term SNP is used to describe what is actually mismatched base due to sequence error. "Insertions" and "Deletions", normally terms used to describe actual genome differences, could be re-termed as "False-insertions" and "False-deletions".
 - b. On Page 4, the term "contracted" is used, whereas the term "collapsed" is used elsewhere in Fig 2C. To me "contracted" sounds like the result of an evolutionary process as opposed to an assembly artifact issue. Perhaps "compression" would be the correct term for an assembly artifact caused by adjacent repetitive regions?
 - c. I would suggest the authors review metrics and terms used in the Assemblython-type publications, and try to improve consistency with existing literature.
- 2) Where is the evidence that base quality is on par with TAIR10 as contended on page 5, "In one ONT MinION run we sequenced and assembled an *A. thaliana* accession (KBS-Mac-74) to a high continuity and base quality on par with the current "gold standard" TAIR10 assembly of the Col-0 reference accession." Can similar metrics, i.e. generation of PCR-free Illumina reads from Col-0 and mapping to TAIR10, be used as a control so as to back up this statement? Alternatively are the authors relying on prior literature to make this statement (in which case should be cited)?
- 3) It would be useful for authors to provide definitions of metrics used in Table S1, such as "split-reads (#)", "paired (%)", "singletons (%)", and "Inter-contig mappings (%)". Why is it that the latter 3 categories do not add up to the total that "Mapped (%)"? For example in Table S1, "ONTmin iteration 0" has 95% mapped short reads, of which 67.7% are paired, 1.8% singleton, and 11.9% on different contigs. What about the remaining ~14%? Are these mapped but in the incorrect orientation or positioned too close or too distant on the same contig? If the latter, what criteria were used?
- 4) Another place that might cause confusion is on page 3, "Therefore, the minimap/miniasm assembly has many local mis-assemblies compared to the Canu assemblies." How can consensus read correction using racon and polishing using short reads fix errors caused by mis-assembly, as opposed to just correcting the incorporated read? Perhaps an illustrative example as a supplemental figure could be used to demonstrate this?

Reviewer #3 (Remarks to the Author):

Summary: In this manuscript, Michael et al. present the results of their reassembly of the Arabidopsis thaliana genome using Oxford Nanopore reads. The authors demonstrate that a single Oxford Nanopore flowcell (along with an undisclosed amount Illumina short reads) is sufficient to generate a highly contiguous reference genome for the species, and they tout the rapidity of their assembly of this reference. While the validation of this assembly and the comparative analysis of assembly methods are both excellent in this manuscript, the authors' presentation of this work and description of exact methods are lacking. I believe that the description of their assembly versions is prone to misinterpretation and they did not describe key details in the methods section of the manuscript. Still, I feel that with suitable revision and care, this could be a highly influential article in the field.

Given that the submitted manuscript lacks line numbers, I have tried to annotate my comments with sentence fragments or section references to provide context. Please add line numbers to the next version of the manuscript to assist the reviewers.

Comments:

It is difficult to judge the main focus of this manuscript, as the authors flirt with presenting their work as a “rapid and inexpensive” reference generation experiment, yet also spend most of their time comparing assembly methods and polishing statistics. I would highly recommend that the authors focus more on the methodology and present a figure that shows their recommended pipeline for reference generation. Also, I recommend that they select one reference genome and present that assembly – consistently – in the text as their comparison point for the experiment.

Please provide a comparison of “N base” content for the best reference assembly against the TAIR10 reference in the results and/or discussions.

There are numerous “citation error” annotations in the text corresponding to incomplete references. Please correct and add the proper citation.

Given its known issues with homopolymers

(<http://www.biorxiv.org/content/biorxiv/early/2017/04/20/128835.full.pdf>) I am surprised to see that Albacore v0.8.4 was used to generate the raw base calls from the MinION output. I would recommend that the authors use a later version of Albacore (v1.1.0+) to recall their raw reads from the output fast5 files from MinION sequencing. Only a single comparison (ie. ONTmin vs ONTcanu without polishing) is needed to demonstrate the effects of better input data here.

Page 3, sentence: “We polished the assemblies using three (3) iterations of racon [15], followed by one (1) round of polishing using Illumina PCR-free paired-end reads with pilon [16] (Table 2)” The use of the term “iteration” here is quite misleading. The first three (true) iterations are sequential racon error corrections followed by one round of Pilon polishing. I would recommend that the authors clearly annotate their polishing steps (ie “racon-based” or “pilon-based”) in the text (or possibly a figure) and refer to assemblies with clearly defined terms (e.g. uncorrected minONT vs polished minONT) to avoid reader confusion. Again, a figure (or updated table) would suffice here.

Page 4, sentence: “Artificially expanded regions occur only in the miniasm assemblies, with the exception of four such regions in the ONTcan raw assembly.” Please rephrase and give counts of artificially expanded regions in each assembly – not just for canu.

Page 4, sentence: “The raw ONTmin assembly (iteration: 0) failed to produce good alignments with optical maps due to high nucleotide errors, causing little overlapping labels.” Was this a consequence of the Albacore base caller version used or the inherent error profile of the reads? The NT.BspQI restriction site has a “T” dinucleotide which may be mis-called by the earlier versions of Albacore. Does NT.BspQI site recovery improve with better basecalling at the raw read level?

Page 5, fragment: “Blast searches with the Col-0 At4g30720 genomic sequence against our KBS-Mac-74 ONTmin assembly recovered two hits (99.3 and 98.2% identity) 10 Mb apart on the same contig” Was the ONTmin assembly used already polished using the “iterative” approach mentioned by the authors? If so, metrics on how long it took to polish this assembly to a useable state will be needed to qualify the adverb “rapidly” used in the earlier section of this paragraph. If this was done with the “iteration 0” assembly, the time it took to generate a useable assembly using commodity hardware would still be of interest to the reader in this section.

Page 5, sentence: "Together this assembly cost under 1,000 USD in consumables and instrument depreciation and took less than a week of actual time." Does this factor in the Illumina reagents and platform (the Illumina sequencing platform is not mentioned in the methods) runtime? How was instrument depreciation estimated if the MinION sequencer is leased to the customer as part of Oxford Nanopore's initial purchasing plans? If you include extensive polishing, how does the technician time affect the overall price point presented.

Page 5, sentence: "While the initial quality of the minimap/miniasm assembly was lower than Canu, several rounds of racon and one round of pilon produced an assembly on par with PacBio in both contiguity and base quality." Why was this the case? Why did racon and pilon polishing improve the miniasm assembly's quality beyond the Canu version?

Page 5, fragment: "While the new PacBio Sequel platform is generating even longer reads..." Please revise to read: "While improvements to the PacBio Sequel platform will enable it to generate longer reads than the RSII platform used in this study..."

Page 8, fragment: "polished using Illumina PCR-free 2x250 bp reads" I cannot find any information on how these reads were generated nor on the statistics of the dataset. Please provide information in the methods to state the Illumina sequencing platform, reagents used, depth of coverage and sample used.

Page 9, <https://github.com/fbemm/onefc-oneasm> Please update this repository with the code and intermediate results as promised.

Page 9, ENA accession: PRJEB21270. Please ensure that this repository is made publically accessible.

Table 2: Were the Illumina reads used to validate the assembly also used in the Pilon polishing stage? If so, the comparison profile of variants for the "fourth" iteration (Pilon correction) is biased by the use of the same data to validate and polish the assembly.

Table 2: Methods proposed by Bickhart et al. (2016) Nat Genetics and Jain et al. (<http://www.biorxiv.org/content/biorxiv/early/2017/04/20/128835.full.pdf>) characterize assembly variant content using a "QV" metric and subsequent ROC analysis (provided by FRCbam). This would be a suitable statistical summary of assembly quality metrics for use in this table.

Reviewer #1 (Remarks to the Author):

The article (Manuscript#: NCOMMS-17-15663-T) entitled “High contiguity *Arabidopsis thaliana* genome assembly with a single nanopore flow cell”, by Todd Michael and collaborators delivers “a fast and cost effective reference assembly for an *Arabidopsis thaliana* accession using the USB-sized Oxford Nanopore MinION sequencer and typical consumer computing hardware (4 Cores, 16Gb RAM)”, aiming at reducing sequencing and computing cost for de novo genome assembly. While the work looks interesting, the methodological/biological insights provided here are of limited novelty.

We apologize if we did not properly present the main goal of our work. This is not about new insights derived from comparing a genome from a non-reference strain with the reference strain, and it is not about new computational tools. The goal is to showcase how even when the aim of a project is “merely” the analysis of a single complex region in the entire genome, rapid and inexpensive genome assembly enabled by the Oxford Nanopore platform provides a clear solution that is easily in reach for any molecular biologist.

Specifically, the authors sequenced an *A. thaliana* accession using the Oxford Nanopore MinION sequencer, and used miniasm/racon pipeline to create assemblies that are comparable to CANU assemblies but at lower computing cost. None of the technologies used here were novel and the authors basically provided testing results for these existing technologies using a newly sequenced *A. thaliana* genome, which however provided very limited biological insights.

Again, the aim of the work was not to provide new biological insights about entire genomes, but to make others aware of what anybody can do now. We resolved the complex structure of the genomic region under a QTL using only benchtop equipment and free software in one week, which even BACs coupled to Sanger sequencing could not resolve in a year's work. This is significant step-change.

In particular, the miniasm/racon pipeline has been shown to work well with *C. elegans* PacBio data (Sović et al. 2017. *Genome Res.* doi:10.1101/gr.214270.116), which has comparable genome complexity with *A. thaliana*. Although the testing by Sović et al. was performed using PacBio data, the MinION data provided here were of similar quality (similar average length, except some longer reads, and similar error rate), which should lead to comparable results by the same assembly pipeline (and indeed the results of the two genomes generated by miniasm/racon looked similar or comparable in assembly statistics).

Our goal was to highlight the opportunity to leverage ONT technology in a lab setting, and not to justify single molecule sequencing. As stated by the reviewer, this has been shown previously. (The cited *C. elegans* work is not particularly appropriate, since there was no clear indication of the quality of the assembled genome.)

Below are some specific concerns regarding the current study.

In Abstract, the authors stated that “because these technologies are not only costly, but also time and compute intensive, it has been unclear how scalable they are”. This is misleading. PacBio SMRT technology has been successfully used in assembling large plant genomes such as maize, sunflower, and also partially helped wheat genome assembly. Even though these assemblies required a large amount of computing resource, the scalability of these methods has been proved. Even though cost is still a big issue, it is not unacceptable.

We agree that the statement was misleading in terms of its relevance to our core message that ONT opens up new opportunities to individual bench scientists. This has been changed.

Please note that genome assembly, even with the reduced cost, is still not intended to be routinely performed in every lab, especially considering the amount of follow-up work such as genome annotations that is required to make the genome sequence useful.

This is exactly our point: Even if one is interested only in a small region of the genome, complete assembly of the entire genome presents the most straightforward solution! We agree that full genome annotation has now become much more demanding and time consuming than the assembly itself, but this is not germane to our work.

Furthermore, the Arabidopsis genome is just a 'tiny' genome in the plant kingdom (and also very small in animal world). A method that is successful for Arabidopsis doesn't automatically extend to large and complex plant genomes. For example, in our preliminary testing with the recently published maize B73 PacBio data, miniasm clashed on a computer with 1 TB memory. To test the scalability of the methods presented here, the authors should at least use a published SMRT dataset of complex genomes such as maize to make sure that the methods are indeed able to work as expected.

We are not advocating for minimap/miniasm per se. We also use Canu and other assemblers when appropriate. We do find that with high quality, long-read Oxford data minimap/miniasm does outperform other assemblers. We also find that it fails with regular (15 kb cut) PacBio data due to the length of the reads. Justifying minimap/miniasm on a larger genome is outside the scope of the current manuscript.

Reviewer #2 (Remarks to the Author):

This research represents a substantive advance in the field of genomic sequencing, in particular demonstrating that high quality assemblies of a higher plant genome can be achieved using a low-cost, long-read technology, and light-weight computational tools. The work demonstrates that the main weakness of long-read technologies, low base accuracy, need not require computationally expensive read-correction steps prior to assembly. Rather, low accuracy can be overcome using consensus correction and polishing steps performed after primary assembly, as claimed by the developers of these tools in prior publications, but demonstrated independently here. By comparing Oxford Nanopore technologies with PacBio, this work also shows that comparable quality can be achieved but at much lower cost. This work also serves as a guiding example of how genome assemblies can and should be evaluated using optical map comparison and short-read mapping. This work is anticipated to have high impact on the research community as a “methods” paper. Although this work provides a use-case to demonstrate the utility of the resulting Arabidopsis strain KBS-Mac-74 reference assembly in detecting the likely genetic basis of a QTL, it falls short of any comprehensive comparative analysis with the Col-0 strain upon which TAIR10 is based. The latter analysis would increase the intellectual merit of this work, but perhaps is being reserved for a separate publication by these authors.

We thank the reviewer for pointing to an outcome that we did not anticipate based on our experience with the assemblers and data types. First, we wanted to show that sparse graph assembly methods, such as minimap/miniasm could be a game changer in terms of time and effort. We also think that our analysis highlights other applications for correction-free assembly. Second, we had been using optical maps for validating contigs and even joining contigs, but the use of the BioNano maps to assess assembly quality was a new application, which we think provides yet another check of assembly quality.

Our goal was primarily to show how an individual bench scientist could use the ONT MinION platform to resolve a complex region of the genome within a week. Therefore, we focused on sequencing the KBS-Mac-74 accession, but have added a new paragraph highlighting identification of structural variants between KBS-Mac-74 and the reference strain Col-0.

The manuscript should be improved in the following ways before publication:

1) More precise terms to describe sequence/assembly artifacts so as to not confuse with real genetic variation:

a. For example, in Table 2 and elsewhere the term SNP is used to describe what is actually mismatched base due to sequence error. “Insertions” and “Deletions”, normally terms used to describe actual genome differences, could be re-termed as “False-insertions” and “False-deletions”.

Good suggestion--done.

b. On Page 4, the term “contracted” is used, whereas the term “collapsed” is used elsewhere in Fig 2C. To me “contracted” sounds like the result of an evolutionary process as opposed to an assembly artifact issue. Perhaps “compression” would be the correct term for an assembly artifact caused by adjacent repetitive regions?

Changed.

c. I would suggest the authors review metrics and terms used in the Assemblython-type publications, and try to improve consistency with existing literature.

We have reviewed our use of terms and tried to make them more consistent with Assemblathon type language.

2) Where is the evidence that base quality is on par with TAIR10 as contended on page 5, “In one ONT MinION run we sequenced and assembled an *A. thaliana* accession (KBS-Mac-74) to a high continuity and base quality on par with the current “gold standard” TAIR10 assembly of the Col-0 reference accession.” Can similar metrics, i.e. generation of PCR-free Illumina reads from Col-0 and mapping to TAIR10, be used as a control so as to back up this statement? Alternatively are the authors relying on prior literature to make this statement (in which case should be cited)?

We assessed the base quality of the TAIR10 reference genome with the same approach that we used for our own assembly, with newly generated Illumina PCRfree data for the reference strain Col-0. Base quality lines were added to figure 2 and the narrative was updated as well.

3) It would be useful for authors to provide definitions of metrics used in Table S1, such as “split-reads (#)”, “paired (%)”, “singletons (%)”, and “Inter-contig mappings (%)”. Why is it that the latter 3 categories do not add up to the total that “Mapped (%)”? For example in Table S1, “ONTmin iteration 0” has 95% mapped short reads, of which 67.7% are paired, 1.8% singleton, and 11.9% on different contigs. What about the remaining ~14%? Are these mapped but in the incorrect orientation or positioned too close or too distant on the same contig? If the latter, what criteria were used?

The mapping statistic categories were previously used to described assembly quality. As pointed out, this approach has various caveats. We adopted the quality assessment method from (Bickhart et al. 2017; Jain et al. 2017) and removed Supplemental Table 1.

4) Another place that might cause confusion is on page 3, “Therefore, the minimap/miniasm assembly has many local mis-assemblies compared to the Canu assemblies.” How can consensus read correction using racon and polishing using short reads fix errors caused by mis-assembly, as opposed to just correcting the incorporated read? Perhaps an illustrative example as a supplemental figure could be used to demonstrate this?

Rephrased make it clearer that the per base quality of the minimap/miniasm is generally lower due to the absence of a read error correction step. Although the reviewer’s comments open up an important questions, we prefer not to discuss the pros and cons of read correction prior to assembly. Of note, the latest developments in genome assembly indicate a paradigm shift towards not correcting reads prior to assembly (<https://dazzlerblog.wordpress.com/2017/04/22/1344>).

Reviewer #3 (Remarks to the Author):

Summary: In this manuscript, Michael et al. present the results of their reassembly of the *Arabidopsis thaliana* genome using Oxford Nanopore reads. The authors demonstrate that a single Oxford Nanopore flowcell (along with an undisclosed amount Illumina short reads) is sufficient to generate a highly contiguous reference genome for the species, and they tout the rapidity of their assembly of this reference. While the validation of this assembly and the comparative analysis of assembly methods are both excellent in this manuscript, the authors' presentation of this work and description of exact methods are lacking. I believe that the description of their assembly versions is prone to misinterpretation and they did not describe key details in the methods section of the manuscript. Still, I feel that with suitable revision and care, this could be a highly influential article in the field.

Thank you for your support of our work.

Given that the submitted manuscript lacks line numbers, I have tried to annotate my comments with sentence fragments or section references to provide context. Please add line numbers to the next version of the manuscript to assist the reviewers.

Done.

Comments:

It is difficult to judge the main focus of this manuscript, as the authors flirt with presenting their work as a “rapid and inexpensive” reference generation experiment, yet also spend most of their time comparing assembly methods and polishing statistics. I would highly recommend that the authors focus more on the methodology and present a figure that shows their recommended pipeline for reference generation. Also, I recommend that they select one reference genome and present that assembly – consistently – in the text as their comparison point for the experiment.

Our main goal was to highlight the ONT minimap/miniasm assembly as a fast method for individual bench scientists, but we thought it is important to demonstrate that the overall assembly is of high quality.

Please provide a comparison of “N base” content for the best reference assembly against the TAIR10 reference in the results and/or discussions.

Our assemblies do not contain Ns since they are contigs and not scaffolds.

There are numerous “citation error” annotations in the text corresponding to incomplete references. Please correct and add the proper citation.

Fixed.

Given its known issues with homopolymers

(<http://www.biorxiv.org/content/biorxiv/early/2017/04/20/128835.full.pdf>) I am surprised to see that Albacore v0.8.4 was used to generate the raw base calls from the MinION output. I would recommend that the authors use a later version of Albacore (v1.1.0+) to recall their raw reads from the output fast5 files from MinION sequencing. Only a single comparison (ie. ONTmin vs ONTcanu without polishing) is needed to demonstrate the effects of better input data here.

We tested this and found that the latest versions of Albacore did not produce improved final results. We have not noted improved homopolymer calling with the updated version of

Albacore in either whole genome sequencing or amplicon sequencing.

Page 3, sentence: "We polished the assemblies using three (3) iterations of racon [15], followed by one (1) round of polishing using Illumina PCR-free paired-end reads with Pilon [16] (Table 2)" The use of the term "iteration" here is quite misleading. The first three (true) iterations are sequential racon error corrections followed by one round of Pilon polishing. I would recommend that the authors clearly annotate their polishing steps (ie "racon-based" or "pilon-based") in the text (or possibly a figure) and refer to assemblies with clearly defined terms (e.g. uncorrected minONT vs polished minONT) to avoid reader confusion. Again, a figure (or updated table) would suffice here.

Excellent suggestion that we have followed.

Page 4, sentence: "Artificially expanded regions occur only in the miniasm assemblies, with the exception of four such regions in the ONTcan raw assembly." Please rephrase and give counts of artificially expanded regions in each assembly – not just for canu.

We have addressed this in the text.

Page 4, sentence: "The raw ONTmin assembly (iteration: 0) failed to produce good alignments with optical maps due to high nucleotide errors, causing little overlapping labels." Was this a consequence of the Albacore base caller version used or the inherent error profile of the reads? The NT.BspQI restriction site has a "T" dinucleotide which may be mis-called by the earlier versions of Albacore. Does NT.BspQI site recovery improve with better basecalling at the raw read level?

We did not note improvements when testing newer version of Albacore. In general, the fact that with several rounds of racon we did not see a problem with the BioNano maps suggests that this is not a systemic base calling problem.

Page 5, fragment: "Blast searches with the Col-0 At4g30720 genomic sequence against our KBS-Mac-74 ONTmin assembly recovered two hits (99.3 and 98.2% identity) 10 Mb apart on the same contig" Was the ONTmin assembly used already polished using the "iterative" approach mentioned by the authors? If so, metrics on how long it took to polish this assembly to a useable state will be needed to qualify the adverb "rapidly" used in the earlier section of this paragraph. If this was done with the "iteration 0" assembly, the time it took to generate a useable assembly using commodity hardware would still be of interest to the reader in this section.

We clarified this in the text. We did use iteration 4, but we also found this using iteration 0 (89% and 85% identity). We do contend that a week is still rapid compared to the weeks/months it took to make fosmids and sequence them in the previous efforts at resolving this QTL.

Page 5, sentence: “Together this assembly cost under 1,000 USD in consumables and instrument depreciation and took less than a week of actual time.” Does this factor in the Illumina reagents and platform (the Illumina sequencing platform is not mentioned in the methods) runtime? How was instrument depreciation estimated if the MinION sequencer is leased to the customer as part of Oxford Nanopore’s initial purchasing plans? If you include extensive polishing, how does the technician time affect the overall price point presented.

We factored in all of these costs into the \$1,000 number. The flowcell cost is \$500 because we buy them in bulk. We have, however, removed any cost related points from the manuscript to avoid any confusion.

Page 5, sentence: “While the initial quality of the minimap/miniasm assembly was lower than Canu, several rounds of racon and one round of pilon produced an assembly on par with PacBio in both contiguity and base quality.” Why was this the case? Why did racon and pilon polishing improve the miniasm assembly’s quality beyond the Canu version?

The unpolished Canu consensus sequences can still contain up to 1% of errors. See the Canu documentation for details (<http://canu.readthedocs.io>; Consensus Accuracy). Thus, a polished minimap/miniasm assembly can exceed the quality of an unpolished Canu assembly. Applied to both, minimap/miniasm and Canu assembly the polishing strategy resulted in a similar quality of all our Oxford Nanopore assemblies as expected. We clarified this in the text.

Page 5, fragment: “While the new PacBio Sequel platform is generating even longer reads...” Please revise to read: “While improvements to the PacBio Sequel platform will enable it to generate longer reads than the RSII platform used in this study...”

We have considered the reviewer’s comment on the latest PacBio Sequel platform developments and instead of just changing the text, replaced the PacBio RSII dataset from the first version of our paper with data from the Sequel platform for the same accession. The text was changed accordingly.

Page 8, fragment: “polished using Illumina PCR-free 2x250 bp reads” I cannot find any information on how these reads were generated nor on the statistics of the dataset. Please provide information in the methods to state the Illumina sequencing platform, reagents used, depth of coverage and sample used.

We have added information about Illumina sequencing to the methods and updated the text where necessary to make it clear that Illumina reads were used for polishing.

Page 9, <https://github.com/fbemm/onefc-oneasm> Please update this repository with the code and intermediate results as promised.

Updated.

Page 9, ENA accession: PRJEB21270. Please ensure that this repository is made publically accessible.

Done.

Table 2: Were the Illumina reads used to validate the assembly also used in the Pilon polishing stage? If so, the comparison profile of variants for the “fourth” iteration (Pilon correction) is biased by the use of the same data to validate and polish the assembly.

Yes, we used the same reads, so we have added a caveat in the text.

Table 2: Methods proposed by Bickhart et al. (2016) Nat Genetics and Jain et al. (<http://www.biorxiv.org/content/biorxiv/early/2017/04/20/128835.full.pdf>) characterize assembly variant content using a “QV” metric and subsequent ROC analysis (provided by FRCBam). This would be a suitable statistical summary of assembly quality metrics for use in this table.

Great suggestion, added to supplement.

REVIEWERS' COMMENTS:

Reviewer #1 (Remarks to the Author):

The revised version of manuscript by Todd P. Michael et al. entitled "High contiguity Arabidopsis thaliana genome assembly with a single nanopore flow cell" has been significantly improved in writing. The work described in the manuscript is good, is well done. However, it still suffers from the lack of novelty (as I commented previously), by comparing it to the other papers published in Nature Communications, due to the previously published work using similar technologies, eg, comparing to the recently published *Solanum pennellii* genome also using Nanopore data (The Plant Cell, 2017 <https://doi.org/10.1105/tpc.17.00521>), and many other genome papers using PacBio SMRT data. Specifically, the technical work described in this manuscript includes Nanopore sequencing of an Arabidopsis genome, comparison of assemblers such as CANU, Falcon, Miniasm/Recon pipeline, and using BioNano maps to improve scaffold size and fix errors. None of these works are novel (due to the fact that so many complex genomes have been assembled by long read sequencing – see examples below and refs cited in the manuscript) except on a new genome. However, the genome described here is too small to give more insights on the usage of these technologies.

The most interesting part of the paper is the last paragraph in Results section about "to support the analysis of two interacting QTL, SG3 and SG3i." Unfortunately, the genes underlying the two QTL have been previously sequenced (in ref 23), and no new casual genes were discovered in the new assembly. Therefore, the major contribution in current manuscript is to have successfully sequenced/assembled the region with many repetitive elements. However, assembly of such regions is very much expected from the long read technologies of PacBio and Nanopore and thus not novel. For example, many gaps in reference genome were closed by PacBio SMRT sequencing in many complex genomes such as human (Nature 538, 243–247, 2016, genome size of ~3 Gb, with a contig N50 size of 17.9 Mb and a scaffold N50 size of 44.8 Mb, resolving 8 chromosomal arms into single scaffolds), maize (Nature 546, 524–527, 2017, genome size of ~2.1 Gb, with max contig length of 7.26 Mb and N50 size of 1.18 Mb), rice (Nature Communications 8, Article number: 15324 (2017), genome size of ~390 Mb, with only 5 gaps on 12 chromosomes) etc. For direct comparison, the recently published *Solanum pennellii* genome has a genome size of ~900 Mb, with max contig length of 12.7 Mb and N50 size of 2.5 Mb. Many complex regions including centromeres/subtelomeres were assembled using PacBio data in other organisms. The large contig N50 size of Arabidopsis (comparing with *Solanum pennellii*) is solely due to its small genome size (and the long read length and the advancement of assembly software) rather than by any novel methods not described previously.

The authors claimed that "The main purpose of demonstrating that highly contiguous genome assemblies can be produced "at the bench," in order to resolve structural variation at a specific QTL." This probably should be changed to "highly contiguous Arabidopsis genome assemblies" – this method cannot be simply applied to large genomes (ie, the cost increase is not in a linear scale relative to genome size). Still, the genome sequencing/assembly (for both cost and quality reasons) currently requires a specialized team, and it is not a regular piece of "bench work". For example, can they produce a much improved *Solanum pennellii* genome (say with a contig N50 size of 5 Mb) "at the bench" using the methods described in this manuscript?

Minor Problems

1. In Abstract "covering 100% (119 Mb) of the non-repetitive genome", evidence is needed to prove this point in Results.
2. "The new KBS-Mac-74 genome was used to resolve a quantitative trait locus that had previously been recalcitrant to a Sanger-based BAC sequencing approach."

This sentence is sort of misleading since it uses "locus" to refer to the whole region (or the repeat part). From ref 23: "Our sequencing results on the BAC identify a 10 kb-region (including At4g30720' paralogue) at SG3i clearly corresponding to the SG3 region surrounding At4g30720." And in current manuscript "SG3 had already been fine-mapped to a 9-kb region around the At4g30720 gene. The A.

thaliana Bur-0 accession has an additional copy of this gene, about 10 Mb away on the same chromosome, and this defines the SG3i QTL." Clearly, the causal gene underlying the QTL was sequenced successfully. Therefore it is the complex repeats in the region that are hard to sequence, but not the genic part. If "an additional copy of this gene defines the QTL", then the repeats do not necessarily belong to the QTL.

Reviewer #2 (Remarks to the Author):

We agree with the comments made by the authors, the focus is technology driven, and the ability for an individual laboratory, to sequence a small size genome, assembly, and validate a region in two weeks is a substantial technological advance.

I recommend moving forward with publication based on the revisions to the manuscript, updates to software repository, and the release of the sequence to GenBank.

Reviewer #3 (Remarks to the Author):

Summary: The authors have resolved many of my previous concerns with their manuscript in this revision; however, several minor points remain.

Line 91: Given their response to my previous question, it sounds as if the authors recalled bases from the Oxford Nanopore dataset using a different version of the Albacore basecaller. A brief description of this effort should be mentioned in the text to provide further information to the reader.

Line 212: Are the "Q" values the "QV" values described in the methods? If so, some additional context is required in the text.

Line 249: More discussion of the influence of polishing on the mapping of BioNano Nt.BspQI sites to assembly versions is needed. Given that many assemblies are subsequently scaffolded using the alignment of secondary datasets (ie. Hi-C, Mate-pairs, Optical maps), the observed improvement of Nt.BspQI site recognition would be useful information to the reader.

For the repository at: <https://github.com/fbemm/onefc-oneasm> , please include a list of required software packages and some more descriptive comments indicating the use of each package in polishing the assembly.

Response to Reviewer's Comments

Reviewer #1 (Remarks to the Author):

The revised version of manuscript by Todd P. Michael et al. entitled “High contiguity *Arabidopsis thaliana* genome assembly with a single nanopore flow cell” has been significantly improved in writing. The work described in the manuscript is good, is well done. However, it still suffers from the lack of novelty (as I commented previously), by comparing it to the other papers published in Nature Communications, due to the previously published work using similar technologies, eg, comparing to the recently published *Solanum pennellii* genome also using Nanopore data (The Plant Cell, 2017 <https://doi.org/10.1105/tpc.17.00521>), and many other genome papers using PacBio SMRT data.

Thank you for the reference to the recently published *Solanum* genome, which only became available during this review process - we were of course aware of the earlier biorxiv submission. This work bolsters the case that Oxford Nanopore MinION work can produce a high quality genome, and we include this now in the introduction. While this reference shows, as others have, that sequencing a larger genome is possible with the Oxford Nanopore MinION, it does however not provide a comprehensive analysis of the quality of the resulting genome. In contrast, the goal of our work was to demonstrate quality of a MinION-derived genome, as well as that it is time to divert activities such as cloning QTLs away from Sanger based applications. The novelty of our work is twofold. First, we do a comprehensive analysis of the quality of the Oxford Nanopore genome using several independent long-molecule methods on a platinum eukaryotic reference (*Arabidopsis thaliana*). Second, we demonstrate that a researcher in the lab could use this approach to go directly to a gene of interest including its wider genomic neighborhood, instead of using other more labor intensive methods that may not actually answer the question at hand. To our knowledge, for a larger eukaryotic genome with some repeat complexity, there is not a reference in the literature that provides researchers with the evidence that Oxford Nanopore performs at the same level as other gold standard long read technologies but with the ease of working at the bench.

Specifically, the technical work described in this manuscript includes Nanopore sequencing of an *Arabidopsis* genome, comparison of assemblers such as CANU, Falcon, Miniasm/Recon pipeline, and using BioNano maps to improve scaffold size and fix errors. None of these works are novel (due to the fact that so many complex genomes have been assembled by long read sequencing – see examples below and refs cited in the manuscript) except on a new genome. However, the genome described here is too small to give more insights on the usage of these technologies.

The goal of the paper is neither to compare assemblers or to create novelty around assembly. As mentioned before, we aimed to demonstrate that a researcher at the bench could use latest long-read technologies to address a specific scientific question and achieve it with high quality results. We also did not use the BioNano maps to improve scaffold size or fix errors, but rather

to assess the quality and contiguity of the assembly, the former being a novel use of the Bionano reads. In our experience, and also those of the community, the tools we describe are all scalable to human or maize size genomes. Generally, new technologies are tested on smaller, inbred genomes and then scaled up to larger ones.

The most interesting part of the paper is the last paragraph in Results section about “to support the analysis of two interacting QTL, SG3 and SG3i.” Unfortunately, the genes underlying the two QTL have been previously sequenced (in ref 23), and no new casual genes were discovered in the new assembly. Therefore, the major contribution in current manuscript is to have successfully sequenced/assembled the region with many repetitive elements. However, assembly of such regions is very much expected from the long read technologies of PacBio and Nanopore and thus not novel. For example, many gaps in reference genome were closed by PacBio SMRT sequencing in many complex genomes such as human (Nature 538, 243–247, 2016, genome size of ~3 Gb, with a contig N50 size of 17.9 Mb and a scaffold N50 size of 44.8 Mb, resolving 8 chromosomal arms into single scaffolds), maize (Nature 546, 524–527, 2017, genome size of ~2.1 Gb, with max contig length of 7.26 Mb and N50 size of 1.18 Mb), rice (Nature Communications 8, Article number: 15324 (2017), genome size of ~390 Mb, with only 5 gaps on 12 chromosomes) etc. For direct comparison, the recently published *Solanum pennellii* genome has a genome size of ~900 Mb, with max contig length of 12.7 Mb and N50 size of 2.5 Mb. Many complex regions including centromeres/subtelomeres were assembled using PacBio data in other organisms. The large contig N50 size of *Arabidopsis* (comparing with *Solanum pennellii*) is solely due to its small genome size (and the long read length and the advancement of assembly software) rather than by any novel methods not described previously.

We agree that PacBio long reads are a well established method in the field and this is exactly why we have included newest Sequel version of the *Arabidopsis thaliana* genome for comparison. Each of the references cited required a great amount of work to identify the structural variation. For instance, in the rice reference a complex fosmid approach is employed along side PacBio reads to attain the high contiguity and resolve very complex repeats. In contrast, the assembly of the complex repeat region around the SG3i QTL was achieved with minimal effort and cost, thus making it accessible to a bench scientist.

The authors claimed that “The main purpose of demonstrating that highly contiguous genome assemblies can be produced “at the bench,” in order to resolve structural variation at a specific QTL.” This probably should be changed to “highly contiguous *Arabidopsis* genome assemblies” – this method cannot be simply applied to large genomes (ie, the cost increase is not in a linear scale relative to genome size). Still, the genome sequencing/assembly (for both cost and quality reasons) currently requires a specialized team, and it is not a regular piece of “bench work”. For example, can they produce a much improved *Solanum pennellii* genome (say with a contig N50 size of 5 Mb) “at the bench” using the methods described in this manuscript?

We recently sequenced a 800 Mb genome in under a week with 4 flow cells (with two MinIONs) and we did use a modest cluster to assemble but this could have also been done with a laptop (over a two week period). The DNA extraction and the sequencing was done by a lab technician (who had some experience sequencing) but now we are training research assistants to run the MinION because it is so easy. In terms of the informatics, a molecular biology graduate student with some experience could handle all the steps required to assemble at least up to 800 Mb. The *Solanum pennellii* is a bit larger but we are fully confident that the methods we describe could be applied. We have found that contig N50 length is primarily limited by the quality and quantity of the HMW DNA, but we don't know of any technical limitation to obtain this result on the bench based on our experience. We looked at the *S. pennellii* sequencing data and they suggest a very tight size selection, but many of the really long reads are missing. This is in contrast to our "no shear approach," which is also a novel aspect of our work that we have not found many people reporting in the Oxford Nanopore community or publishing on. We have added a few sentences in the Results, Methods and Discussion to clarify that we used this technique.

Minor Problems

1. In Abstract "covering 100% (119 Mb) of the non-repetitive genome", evidence is needed to prove this point in Results.

Removed statement in the abstract to shorten the abstract.

2. "The new KBS-Mac-74 genome was used to resolve a quantitative trait locus that had previously been recalcitrant to a Sanger-based BAC sequencing approach."
This sentence is sort of misleading since it uses "locus" to refer to the whole region (or the repeat part). From ref 23: "Our sequencing results on the BAC identify a 10 kb-region (including At4g30720' paralogue) at SG3i clearly corresponding to the SG3 region surrounding At4g30720." And in current manuscript "SG3 had already been fine-mapped to a 9-kb region around the At4g30720 gene. The *A. thaliana* Bur-0 accession has an additional copy of this gene, about 10 Mb away on the same chromosome, and this defines the SG3i QTL." Clearly, the causal gene underlying the QTL was sequenced successfully. Therefore it is the complex repeats in the region that are hard to sequence, but not the genic part. If "an additional copy of this gene defines the QTL", then the repeats do not necessarily belong to the QTL.

Knowing that a functional paralog maps at the duplicate locus (that's all we knew after ref 23) isn't necessarily enough to understand biology and evolution of the underlying trait. For instance, in another case of duplicated genes absent from the reference Columbia (<https://www.ncbi.nlm.nih.gov/pubmed/22285031>), the paralog was functional (expressed) only because the extra copy had been inserted behind a promoting region, which was not sensitive to the RNA directed DNA methylation mechanism, which in turn is induced at another complex structural rearrangement of the same gene (multiple inverted-repeat copies). Hence, the whole structural variation at the duplicate locus has potential to be involved in the trait mechanism. Indeed, 'locus' = the genomic region and its involvement in a specific QTL doesn't necessarily

lie (only) in a gene that it contains. Knowing its complete sequence (and not only that it contains a specific gene) definitely helps (regulatory regions).

Reviewer #2 (Remarks to the Author):

We agree with the comments made by the authors, the focus is technology driven, and the ability for an individual laboratory, to sequence a small size genome, assembly, and validate a region in two weeks is a substantial technological advance.

I recommend moving forward with publication based on the revisions to the manuscript, updates to software repository, and the release of the sequence to GenBank.

Reviewer #3 (Remarks to the Author):

Summary: The authors have resolved many of my previous concerns with their manuscript in this revision; however, several minor points remain.

Line 91: Given their response to my previous question, it sounds as if the authors recalled bases from the Oxford Nanopore dataset using a different version of the Albacore basecaller. A brief description of this effort should be mentioned in the text to provide further information to the reader.

Recently a neat comparison of the current Oxford Nanopore basecallers was released online (<https://github.com/rrwick/Basecalling-comparison>). The “Read Identity” and “Assembly Identity” graphs are consistent with our findings that there are only moderate improvements in identity over the evolution of the Oxford Albacore basecaller during the period of time that this manuscript has been under review. We have added a brief comment in the results and discussion concerning the evolution of the basecaller. We don’t think citing the github paper is appropriate since it may not be maintained.

Added to the results section concerning basecalling evolution:

“Oxford Nanopore is rapidly improving the Albacore base-caller and subsequent versions (up to v2.1.3 in December 2017) provided only minimal quality and quantity improvements. The modest improvements are consistent with other ONT plant sequencing projects¹⁴ and may reflect that the Albacore RNN was not trained on unamplified plant DNA.”

Added to the methods section concerning basecalling evolution:

“Oxford Nanopore is rapidly developing the Albacore RNN basecaller and several version of Albacore were released during the review of this work (v0.8.4 to v2.1.3). We tested v1.2.1, v1.2.4, v2.0.2 and v2.1.3 and found that the quality and quantity of sequence called was only marginally better than then v0.8.4 consistent with another report on basecalling unamplified plant DNA¹⁴”

Line 212: Are the “Q” values the “QV” values described in the methods? If so, some additional context is required in the text.

Q values are indeed QV values. We corrected the manuscript and added a pointer to the corresponding method section that explains the concept.

Line 249: More discussion of the influence of polishing on the mapping of BioNano Nt.BspQI sites to assembly versions is needed. Given that many assemblies are subsequently scaffolded using the alignment of secondary datasets (ie. Hi-C, Mate-pairs, Optical maps), the observed improvement of Nt.BspQI site recognition would be useful information to the reader.

Great suggestion - we have addressed this in a revised paragraph in the Discussion.

For the repository at: <https://github.com/fbemm/onefc-oneasm> , please include a list of required software packages and some more descriptive comments indicating the use of each package in polishing the assembly.

We added a list of software package to each section of the github wiki. Headings were extended to be more descriptive and the used software packages are now indicated at each step